# Theoretical Exploration of Supply Chain Viability Utilizing Blockchain Technology

**Weili Yin**  **and Wenxue Ran ***

Logistics School, Yunnan University of Finance and Economics, Kunming 650221, China; yinweili12345@163.com
* Correspondence: ranwxa@vip.sina.com

**Abstract:** As a disruptive and innovative technology, blockchain will significantly revolutionize how organizations produce and operate as global rivalry intensifies. The global COVID-19 outbreak, combined with the growing complexity of supply chain networks, has exposed supply chains' vulnerability to disruption. Therefore, improving the supply chain viability is the primary way to deal with the risk of supply chain disruption. Using the method of literature research, this conceptual paper systematically reviewed and sorted out relevant literature, extracted corresponding capabilities, and put forward relevant research propositions. From the perspective of the resource-based view and resource-dependent theory, this study investigates specific dimensions of the blockchain-enabled supply chain capability: connectivity, network capability, and supply chain reconfiguration and the impact of external resource-dependent capability on the viability of the supply chain. The propositions show that the blockchain-enabled supply chain capability, and external resource-dependent capability will positively impact supply chain viability. It is expected to assist supply chain firms in implementing blockchain technology to increase supply chain viability and improve their capacity to achieve sustainable supply chain development during the crisis.

**Keywords:** blockchain technology; supply chain viability; resource dependence theory; resource based view; TOE theory

## 1. Introduction

As the most disruptive technology, blockchain was introduced in 2008 as a concept of Bitcoin which was originally proposed anonymously by Nakamoto [1]. As the bitcoin trend grew, researchers focused on the blockchain technology behind Bitcoin, and some scholars have suggested that blockchain technology could be applied to supply chain management [2].

In particular, the worldwide pandemic has caused unprecedented damage to the global supply chain. To meet customer needs, how enterprises can improve their supply chains to build supply chain viability and promote the sustainable development of their supply chains has become an essential task [3].

Technology, especially digital information technology, can provide timely, efficient, transparent and heavily resilient information on the supply chain [4]. The emerging generation blockchain technology can support the practice of circular economy but also can help enterprises establish resilience. As enterprises focus more on cost savings, the supply chain's vulnerability became more pronounced during the global COVID-19 pandemic, resulting in severe disruptions to the supply chain of many enterprises [5]. At the same time, enterprises are facing an increasingly dynamic environment. The complexity of global supply chain networks [6], making it difficult for organizations to track products, data information and evaluate data information effectively while leveraging features such as the reliability, traceability, immutable, and innovative contracts of blockchain technology can make enterprises decrease the demand for intermediaries [7] and thus improve the resilience and resilience of the supply chain [8].

Blockchain technology can build localized, agile, and digital supply chains to help organizations increase their supply chain resilience [9]. Scholars believe that the lack of trust between supply chain disruption partners is a significant issue affecting transactions because collaboration among members requires sharing data information to improve supply chain disruption visibility [10]. Blockchain technology allows partners to share information in a securely and transparently manner and rapidly trust partners while enhancing cooperation [11].

Viability is the supply chain's (SC) ability to sustain itself and survive in a changing environment by redesigning its structure and re-planning performance for long-term impact. Viability is considered the fundamental attribute of SC and spans three perspectives: agility, resilience, and sustainability. Supply chain viability adapts to positive changes (e.g., agility perspectives), absorbs harmful interference, and recovers and survives short-term and long-term outages.

This research phenomenon is of great practical significance to study the supply chain viability of blockchain technology. Blockchain as a new technology, technology is essentially the resources of enterprises, and the purpose of enterprises is to convert resources into enterprise capabilities, so this study based on the concept of the resource-based view and resource dependence theory to explore how to use blockchain technology to improve the viability of the supply chain? According to the resource-based view, the firm's internal capabilities are conceptualized, and according to the resource-dependent theory, the external capabilities are conceptualized. Therefore, this study explores the following questions:

1.  Which theories are best fit to apply to blockchain research in organizations.
2.  According to the resource-based view, what is the blockchain-enabled supply chain capability and its subdimensions, and based on the resource dependence theory, what are the external capabilities of the enterprise and its subdimensions.
3.  Whether the blockchain technology-enabled supply chain capabilities and external resource capabilities can improve the supply chain viability.

This paper presents a conceptual study, and we propose research topics mainly by reviewing the literature and combining them with observed phenomena. The main contribution of this paper lies in that we identify specific dimensions of the blockchain-enabled supply chain capability and external capability, and we expand the application fields of the resource-based view and resource-dependent theory. We put forth four propositions about blockchain technology and supply chain viability.

The paper is organized as follows: In Section 2, we mainly review the literature. In Section 3, we present the specific blockchain-supported supply chain capabilities from the RBV and RDT perspectives. In Section 4, we demonstrate the relationship between the specific capability and supply chain viability. Finally, Section 5 details the conclusions, which include theoretical and practical contributions.

## 2. Literature Review

### 2.1. Blockchain Technology

Blockchain is the original core technology of Bitcoin, and blockchain technology is a decentralized transaction and data management technology, which is most commonly used in cryptocurrencies such as Bitcoin [1]. The essential feature of blockchain is a shared database in which data or information is stored, with characteristics such as immutability, traceability, transparency, collective maintenance, and so on [12]. Blockchain is a new application model of computer technology such as distributed data storage, point-to-point transmission, consensus mechanism, and encryption algorithms. Every transaction in the blockchain needs to be consistently validated by system participants for traceability and security of information. In contrast, each block in the blockchain is connected to the previous block, making it difficult to modify the information without network consensus [13,14]. Distributed ledgers have the potential to be highly transparent, secure, immutable, and decentralized characteristics. In addition to blockchain technology in financial transactions, these characteristics of blockchain can also be applied in operational and business issues. It

has expanded from supply chain management to hospital records, transparent voting, and digital signatures [15,16].

Blockchain can be divided into three types: public chain, consortium blockchain, and private chain. Blockchain has evolved over the decades and can be divided into four stages depending on the different types of blockchain technology and usage characteristics: The blockchain 1.0 phase focused on trading, mainly using the dense currency [17], notably Bitcoin. This decentralized digital currency enables peer-to-peer transactions through cryptography. Blockchain 2.0 is an extension of blockchain 1.0, a phase in which key features include privacy, smart contracts, and the emergence of blockchain tokens and features for non-local assets. The most well-known platform at this stage is the emergence of Ethereum, but there are many other solutions. Blockchain 3.0 further expands the focus of blockchain to more complex smart contracts that extend blockchain to all aspects of human life outside finance. The blockchain 4.0 phase incorporates artificial intelligence technology into blockchain technology [18]. The development stages of blockchain are shown in Figure 1.

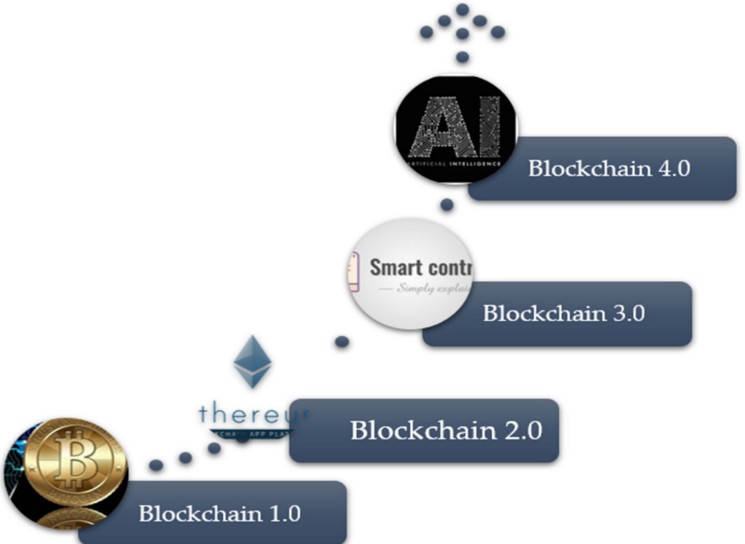

**Figure 1.** The development stages of blockchain.

The understanding of blockchain technology can be studied from three perspectives: Firstly, we can regard blockchain as a distributed ledger technology that can avoid centralized management of the ledger brought about by various security risks, trust risks, as individuals cannot have control over the ledger; Secondly, we regard blockchain technology as a combination of technologies, just as many innovations are based on different combinations, and are not entirely disruptive 0–1 innovations, blockchain-related technologies such as asymmetric encryption, peer-to-peer (P2P) technology, timestamps, smart contracts, consensus algorithms and other technologies, are not entirely disruptive innovations, and blockchain choose different functions based on the different needs of a particular scene. Thirdly, we regard blockchain as a general universal technology, like cloud computing, the internet of things, big data, and other technologies as a standard technology. Blockchain technology is characteristic of treating blockchain technology as a general digital infrastructure that enables it to support our lives and change our way of life like the internet. Blockchain differs from the traditional internet in that it can only transmit and store information; it also allows the value of digital assets to flow efficiently over the internet, allowing money to be treated as information.

*2.2. Blockchain-Based Supply Chain Management Research*

From the perspective of endogenous economic growth theory, technology is the endogenous driving force to maintain economic growth alongside capital and labor. In

the same way, blockchain also has this potential. The application of blockchain technology has been derived from financing, electronic certificates, supply chain management, e-government, and other fields [19].

Many previous studies focus on blockchain technology, and most of the current research on blockchain concentrates on the use of blockchain, blockchain technology, research on the effects of blockchain on enterprise performance, etc. Dutta et al. [20] examined the integration of blockchain technology in supply chain operations, highlighting opportunities, possible social impacts, current state-of-the-art technologies, and critical research trends and challenges. Cui and Idota [21] proposed the concept of supply chain information system reorganization based on blockchain technology, aiming at the pain points of low trust and untimely information exchange in the traditional supply chain information system. Morkunas et al. [22] examined how the use of different types of blockchain technology affects business models and outlines the impact of blockchain technology on each element of the business model. Zavolokina et al. [23], using blockchain alliances as a research object, identified six tensions faced by alliances in the three areas of alliance management, business value, and governance, and ways were proposed to address those tensions to help organizations benefit from the cooperation of blockchain alliances. Saberi et al. [12] have studied how to solve issues related to SC and help the supply chain achieve sustainability with blockchain and they describe four barriers that affect the adoption of blockchain technologies: inter-organizational barriers, intra-organizational barriers, technical barriers and external barriers, while proposing that the transformation of truly blockchain-led businesses and supply chains is still in progress and at an early stage. Min [8] proposed a blockchain architecture and potential remedies for overcoming blockchain challenges, and managers can leverage blockchain technology from a security perspective to improve supply chain resilience. Cole et al. [24] have encouraged research into blockchain technologies from an operational and supply chain management perspective and identify impacts on operational and supply chain management areas through interpretation and analysis of blockchain technologies, and potential areas of research for the future. Van Hoek [25] drew on the radio frequency identification (RFID) research framework to explore how blockchain technology can be implemented in the supply chain and developed a research framework that can provide some reference for management decisions. Queiroz and Fosso Wamba [26] studied some of the factors used in blockchain and, at the individual level, developed a conceptual model based on the classic technology acceptance model and the Unified Theory of Acceptance and Use of Technology (UTAUT) model, drawing on the theories of blockchain, supply chain, network theory, and technology acceptance model, which confirmed the apparent adoption behavior between Indian and U.S. professionals and contributed to the adoption of IT, the supply chain, and blockchain approaches. Treiblmaier [27] has tried to fill the research gap on the potential impact of blockchain on supply chain by using four economic theories: principal-agent theory, transaction cost analysis, resource-based view, and network theory, and explains how to study the impact of blockchain on supply chain management from different perspectives according to different research questions.

Although the adoption, sustainability, and supply chain management of blockchain in the supply chain have been extensively studied in the prior literature, the differences in this study are: First, we use blockchain as an IT technology, from a capability perspective, the blockchain-supported supply chain capability concept into connectivity, network capabilities, and supply chain reconfiguration capabilities. Secondly, we use the concepts of the resource-based view and resource dependence theory to study how internal and external capabilities work together to survive in the supply chain. Finally, we study how blockchain technology can be used to improve the viability of the supply chain.

### 2.3. Supply Chain Viability

Supply chain viability is a new supply chain management theory where viability is defined as the ability of the supply chain to survive in a changing environment. Supply chains are the backbone of the economy and society, and the interactions between supply

chains are becoming more and more complex, presenting distinct characteristics, while supply chains, as a complex system, are very similar in composition to ecosystems. In particular, the vulnerability, resilience, and sustainability of supply chains have been significantly challenged since the outbreak of the COVID-19 pandemic. The global COVID-19 pandemic has greatly influenced economic and social fields and provided a series of new decision-making environments for supply chain research [28]. There will also be unsupportable supply situations for some emergencies, such as the demand surge for masks, disinfectants and protective clothing during the COVID-19 outbreak. Therefore, new environmental conditions also raise new market and social viability issues in the supply chain, where for some industries the sharp decline in demand and supply led to sudden shutdowns, and for some industries because of the new environment new demand was triggered. If the supply chain does collapse, the problem is no longer how to bounce back and return to a normal state but to consider how to adapt and survive under rapidly changing internal and external conditions, which is clearly beyond the scope of supply chain elasticity, agility, flexibility, etc., so the concept of supply chain's viability is raised. Therefore, especially under the influence of COVID-19, supply chain viability has become a very important issue for the supply chain [28].

Viability as long-term viability-oriented viability enables the supply chain to meet viability needs in a changing environment. In summary, supply chain viability refers to a value-added network that adapts dynamically, where structures can change, including the ability to respond quickly to positive changes; the ability to absorb adverse events elastically and have the ability to recover after an outage; the ability to survive in the long term and to ensure that goods and services are delivered by society and markets in response to internal and external changes, in response in turn to sustainable development needs [29].

Supply chain viability capabilities cover the various management and organizational principles of the system, information, and network theory, and the first to propose a viability model was Beer [30]. Beer's model allows us to understand how interrelated operations communicate with changing market environments and metasystems (markets, policies, and societies), and Beer's viability model is based on analogies with human organisms, which are considered to be the most advanced, viability-oriented complex systems, with a viability supply chain as an open system and an open system that interacts with the environment [31], the main features of open systems are control, adaptive, and self-organizing [32].

For example, in times of economic stability, supply chain viability emphasizes making full use of global sourcing, lean production, and agility to provide a wide range of products to meet customers' individual needs. In contrast, in the event of natural disasters, fires, and strikes, supply chain viability emphasizes maintaining supply chain resilience and mitigating risks through the resilient build-up of proactive response capabilities such as capacity flexibility and backup supply chains [33]; In a long-term global disruption such as the global COVID-19 pandemic, supply chain viability emphasizes the transformation of production, reduces product categories, and radically transforms the supplier base and logistics, and local production. The primary role of supply chain viability is to establish, train and implement adaptation and recovery mechanisms [34]. This has an extreme guiding significance for enterprise virtual design and simulation of elasticity and viability supply chain structure, focusing mainly on adaptive training to achieve supply chain viability capacity. Supply chain viability considers both positive events (market growth, profitability) and negative events (supply chain disruptions), which experience positive and negative impacts throughout the life cycle. The main idea of supply chain viability is to provide supply and demand matching multi-structure supply chain design, establish and control the adaptation mechanism between structural designs [34]. Supply chain viability can be profitable in economic stability, withstand supply chain risks and supply chain disruptions, and survive long-term global disruptions caused by enormous economic and social shocks.

Supply chains need to be designed and managed not only to be efficient and resilient but also to be viable so that it can continue to operate and meet the demand in the face of severe damage, and the global COVID-19 pandemic revealed many examples of supply chain lack of viability, complex supply networks due to local node interruption, bullwhip effects, and other factors resulting in loss of connectivity. As a new context for post-epidemic decision-making, the supply chain faces a long-term crisis of uncertainty, and in a pandemic situation, the supply chain does not always allow the supply chain to be resilient. However, if the supply chain is adaptable enough to plan and deploy dynamically according to its circumstances in a supply chain outage, the only way for the supply chain to survive is adaption [28]. Supply chain viability capability is from the perspective of overall adaptation. It extends the concept of supply chain resilience which is a closed system. The viability supply chain is an open system concept. It combines the "bounce-forward-adaptation" choice.

According to the research of Ivanov [29], supply chains are divided into three dimensions: multi-structural viability perspective, viability supply chain model, and supply chain ecosystem perspective. Viability is the ability of a system to maintain itself and recover from long-term interference. In an SC environment, viability can be considered an intersection of elasticity, adaptability, and sustainability.

Through the review and analysis of the literature mentioned above, we find that supply chain viability is essential for the long-term development of enterprises. The study of supply chain viability goes beyond the fields of agility, elasticity, and sustainability in the field of the supply chain, and new problems in practice urgently need new theories to explain the phenomenon of practice. We also found that there is no research about digital technology and emerging technology on the supply chain viability ability related research. Therefore, this study is designed to explore how to use blockchain technology to enhance the viability of the supply chain. On the one hand, it can enrich the emerging supply chain viability of related research. On the other hand, it can also enrich how to use digital technology to enhance the viability of the supply chain the ability to respond flexibly to interference. With the changing environment and society and the rapid development of technology, it is essential to observe and analyze the changing conditions and influence on the supply chain and the progress of digitalization. Continuous reassessment and reassessment of decisions and actions are inevitable to adapt and reprioritize supply chain dynamics.

## 3. Methods

In order to develop the conceptual framework, we first reviewed the extant literature related to blockchain technology and supply chain. This review revealed several types of research related to blockchain technology which focuses on the finance service, electronic deposit certificates, electronic government. Blockchain has shown a wide range of application value in the supply chain field. No matter the government, industry, or major enterprises are actively promoting the innovation and application of blockchain in the supply chain, which has attracted significant attention from academia and industry. However, compared with the rapidly developing blockchain industry, the current research on the supply chain management theory based on blockchain technology is still in its infancy and lacks systematic research. In the face of this enabling of blockchain technology, it is necessary to systematically review and summarize the development process of supply chain management research.

This section mainly uses the Web of Science database, Google Scholar, Emerald, Elsevier, and other databases to search relevant papers, related to blockchain and supply chain viability. These database and search engine was queried for articles and reviews written in English that were published to date inclusive and contain in their title the terms "supply chain", "blockchain", and "supply chain viability". By sorting out these papers, we find that only four articles introduced supply chain viability, which shows detail in

Table 1. Furthermore we did not find the relevant articles investigating the relationship between blockchain and supply chain viability.

**Table 1.** The paper related to supply chain viability.

| Authors (Year) [Reference] | Title | Journal |
|---|---|---|
| Ivanov (2020) [29] | Viable supply chain model: integrating agility, resilience and sustainability perspectives—lessons from and thinking beyond the COVID-19 pandemic | *Annals of Operations Research* |
| Ruel, El Baz, Ivanov and Das (2021) [35] | Supply chain viability: conceptualization, measurement, and nomological validation | *Annals of Operations Research* |
| Ivanov and Dolgui (2020) [36] | Viability of intertwined supply networks: extending the supply chain resilience angles towards survivability. A position paper motivated by COVID-19 outbreak | *International Journal of Production Research* |
| Ivanov (2021) [28] | Supply Chain Viability and the COVID-19 pandemic: a conceptual and formal generalisation of four major adaptation strategies | *International Journal of Production Research* |

Through the review of these four pieces of literature, we find that Ivanov is the most influential scholar in the field of supply chain viability. His research mainly focuses on supply chain resilience and risk management, simulation and data-driven digital supply chain twins. His papers are mainly published in authoritative journals, such as *Transportation Research Part E, International Journal of Production Research, European Journal of Operational Research, Annals of Operations Research* and *Production Planning & Control*. He is also a very influential scholar. He has influenced many famous scholars, as shown in Figure 2.

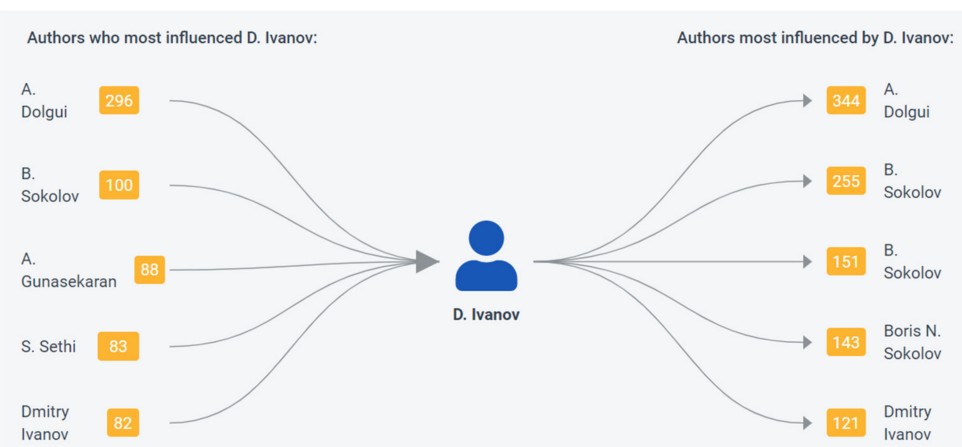

**Figure 2.** Authors who most influenced D. Ivanov.

Even though supply chain management research is well developed, the following issues still exist: Firstly, the supply chain needs multi-party cooperation and lacks efficiency; Second, the supply chain is long and lack trust; Finally, there is the risk of data tampering, and the supervision cannot be completely centralized.

Through a review of the literature on supply chain viability, we find that there has been little research on the relationship between digital technology and supply chain viability, so this paper mainly studies the use of blockchain technology to solve the problems of information sharing and trust in the supply chain and information being easily tampered, and use blockchain technology to improve supply chain viability.

## 4. Blockchain-Supported Supply Chain Capabilities from the RBV and RDT Perspectives

*4.1. Blockchain Enabled the Internal Organizational Capabilities of the Supply Chain*

According to the view of the resource-based view, the resources that an enterprise already owns or may have as a capacity to build competitive advantage, and provides a test of resource relationships and capabilities within an organization, and explains that some organizations are superior to others to gain a competitive advantage [37].

Blockchain technical support supply chain capabilities are expected to provide unique and valuable resources for enterprises, information communication technology (ICT, and knowledge resources that do not automatically improve business performance [38], but the ability to translate the resources that an enterprise has into an enterprise. ICT resources facilitate the creation of capabilities at the enterprise and supply chain levels to create value for enterprises based on their unique value and non-imitation [39]. Therefore, in this context, we take the enterprise's view of resource-based view as a theoretical perspective, this answer for our RQ1, and study whether the ability of the internal organization ability, blockchain technical enabled supply chain, can improve the ability of supply chain viability. We divide the blockchain-enabled supply chain capabilities into three dimensions: connectivity, network capabilities, and supply chain reconfiguration capabilities, this answer for our RQ2, as shown in Figure 3.

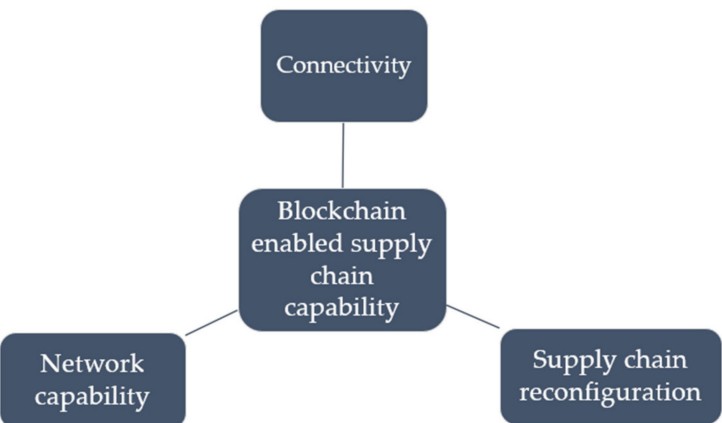

**Figure 3.** The specific dimensions of blockchain-enabled supply chain capability.

### 4.1.1. Connectivity

Connectivity is the essential fit between human society and the internet, and the connection is the basis of various relationships [40]. In the internet era, human society has realized the network-based connection. The combination of the internet as a series of interconnected nodes can effectively meet the human society to achieve the demand for connectivity, which is one of the reasons for human society to socialize the network. While the traditional internet can only achieve the transmission and storage of information, and the use of blockchain technology can be achieved as now processing information to process funds, as easy as sending mail to send funds, accept funds [41] blockchain technology combined with distributed network technology, can address multi-center and multi-intermediary situations, distributed network technology makes each different node through the connection to achieve data information storage [21]. The most important thing when data is transferred synchronously for storage is the connection function. In Chery commercial vehicles, by introducing blockchain and artificial intelligence & internet of things (AIoT) technology for ant chains, the chain service of blockchain is embedded into the vehicle to realize the "survival is chain" of traffic data so that each vehicle becomes a trusted data source. Compared with the vehicle manufacturer providing data records to the blockchain platform, directly entering the vehicle system is more reliable.

For a successful supply chain, organizations often need dynamic and tight structures to monitor material flow, transportation, and potential risks, and in this case, extracting

information through connections and automation without human intervention is a challenge [42]. Therefore, the partially integrated system [43,44] and limited connections [42], cannot realize the desired result.

It is important to note that the implementation of blockchain technology in the supply chain requires different demands for digital interactions from company to company, which requires prior analysis [9,45]. Blockchain technology as a tool provides reliable information and information and visualization for participants in the supply chain, depending on the interaction between digital media. The suitability of the internet of things (IoT) and RFID technologies improve connectivity within and outside the organization to obtain valuable data information. When these technologies are connected to the blockchain platform, they can obtain and disseminate information [46–48].

At present, there is relatively little practical research on blockchain technology applications and blockchain applications in the supply chain [24,49], especially how enterprises connect and formalize operations, which limits the understanding of this technical possibility, as connectivity creates the ability to exchange information and improves business performance [50]. Therefore, it is essential to consider the necessary connections, as this involves the technical requirements and the relationships created in the supply chain, and the functional aspects required by the integration process and its agents. However, blockchain connectivity in the supply chain needs to be based on assessing its needs [51,52] and needs to identify technical needs to supply chain agents, and businesses need to lead and coordinate needs and strengthen business linkages [53].

Based on the view that blockchain connectivity in the supply chain depends on technical interoperability and organizational interoperability, connectivity is essential for the use of blockchain in the supply chain [54] in response to organizationally defined features. The results of using technology depend on what enterprises expect from technology, and on interactions and relationships among members of the supply chain [55]. In this sense, supply chain viability depends on the impact of technical interoperability and organizational interoperability. Interoperability of technology and organization is found to help reduce the need for redundancy in the supply chain [35,56].

Systems between organizations are built around information technology to create, store, convert, and transmit information [57] and the [58]. The core idea of blockchain technology is that both parties can trade directly without trusted third parties, which is also a vital goal of the inter-organization system. Advanced information technology has dramatically improved the company's connectivity, making seamless connectivity possible for individuals anywhere in the chain [59], and enhanced connectivity can significantly increase competitiveness and shorten the development cycle of new products. Real-time connectivity helps organizations respond to feedback from members of the chain promptly, and enhanced connectivity helps make better decisions.

While enterprises have aware of the importance of connectivity, some organizations are reluctant to share information about their businesses because they believe that the flow of information between organizations involves information security issues, especially for some healthcare, utilities, and government departments [60]. Encryption tools that use blockchain technology in supply chain systems are expected to address security issues in transmitting information [61]. As a result, blockchain technology plays a significant role in ensuring the security, availability, and accessibility of the information needed in the supply chain process [62]. Blockchain technology enables the safe flow of information between chains. Blockchain technology-enabled connectivity can promote information sharing between enterprises to improve business performance. Implementing blockchain technology in the supply chain helps organizations achieve end-to-end visibility, which is a considerable challenge for supply chains that do not currently implement blockchain. The reasonable cost of building a blockchain infrastructure is another driver of the active participation of supply chain entities in information sharing [63].

In traditional supply chain scenarios, distributed ledger technology enables the supply chain entities upstream and downstream of the blockchain to access, record, validate and

update the entire transaction database independently, without reconciling the transactions between the transaction subjects [62,64]. Because blockchain technology guarantees transparency and real-time information responsiveness in the supply chain, enterprises can include all transaction entities in their supply chains. In particular, in the event of supply chain disruption, connectivity is even more important, and supply chain capabilities supported by blockchain enable the supply chain to obtain up-to-date information in a short period, thereby adjusting the company's products and services.

### 4.1.2. Network Capabilities

The network capability of an enterprise refers to the ability of an enterprise to build, process, and utilize networks [65]. These capabilities and other enterprise capabilities are intertwined to increase the enterprise's competitive advantage [66]. Network capabilities are dynamic capabilities [67]. Network capabilities enable enterprises to access different resources to identify opportunities and respond quickly to market changes. Good networking capabilities are a vital driver of the success of start-up SMEs [68,69]. In addition, effective management of internal and external information flows helps identify opportunities for innovation [70] and enhances enterprise resilience. At the same time, coordinating internal and external knowledge enables enterprises to quickly identify market trends and respond quickly to market demands [67], so the higher the network capability of an enterprise, the more resilient the supply chain will be.

Network capabilities are developed by interacting with other participants in the network [71] and network capabilities refer to the ability of an enterprise to develop and leverage inter-organizational relationships to obtain resources held by other actors [72] to enable businesses to make better use of an increasingly diverse network [65]. A network-capable enterprise can improve its overall position in the network and develop the ability to manage inter-organizational relationships [73]. Network-capable businesses can build and maintain attractive and influential relationships with external actors. Network-capable enterprises tend to be able to gather more information about competitors [74], By developing network capabilities, organizations can improve internal communication and understanding among partners [75], which helps organizations build trust and a positive communication environment between organizations, and by building network interactions, they can build trusted functional relationships [72].

Network capabilities are dynamic capabilities that generate interdependencies [67]. Network capabilities enable enterprises to access different resources, identify opportunities, and respond quickly to market demands [76]. For SMEs, developed network capabilities are a vital driver of the success of the central enterprise due to limited resources [68] managing internal and external information flows to facilitate knowledge sharing, reduce costs, and more. Practical internal and external information flows help identify opportunities and accelerate innovation [67,70].

In this regard, Walter et al. [72] regard network capability as "*the ability of a company to develop and utilize inter-organizational relationships to obtain resources held by other actors*". The nature of network capabilities is developmental rather than inherent [77]. As a technical tool, blockchain technology can increase connections within and between enterprises and facilitate the sharing of information, which is very important to enhance network capabilities. The use of blockchain technology does not mean that the supply chain is vertically integrated in the traditional sense, but with blockchain technology, the flow of information is more fluid. Blockchain technical support supply chain network capabilities for supply chain enterprises to use blockchain technology to identify external network value and opportunities and help to obtain knowledge and other resources in the network. Network capabilities help organizations identify opportunities for external collaboration, bring together the necessary setup to fully capture knowledge and information in the network to improve their ability to respond to problems [78], improve supply chain resilience to risk, and enhance supply chain viability.

As a disruptive technology, blockchain technology can improve the transparency of information in the supply chain and facilitate the exchange and sharing of information. From the point of view of blockchain technology characteristics, its network capabilities are generated with the emergence of technology, different systems through the network connection, exchange of data and information, the implementation of blockchain technology in the supply chain can promote information sharing, strengthen the connection between different information systems and equipment, and control the source of information effectively. When the supply chain suffers serious interruption, the network capability of the supply chain must be improved to improve the viability of the supply chain. Network capability is a dynamic ability. The networking ability can enable enterprises to obtain more resources, identify market opportunities, and actively respond to market changes [76].

### 4.1.3. Supply Chain Reconfiguration

The supply chain is a complex and dynamic network system that will change the supply chain's size, shape, and configuration over time [79,80]. Changes in the supply chain structure lead to managing and optimizing supply chain processes [34]. The emergence of new technologies is driving changes in the structure of the co-supply chain to enable it to adapt to new technologies and innovations. In severely disruptive situations, which may result in suppliers not being able to supply on time for a short period, the structure of the supply chain must therefore also be redesigned. In particular, the COVID-19 pandemic led to dramatic changes in the global supply network. For supply chain structures, positive technologies and paradigms will produce positive changes to supply chain structure, while disruption risks can have an adverse change in the supply chain structure.

Supply chain structure dynamics occur at the strategic, tactical, and business decision-making levels. At the strategic level, the main competitive advantage of supply chain firms depends mainly on the adoption of new disruptive technologies, supply chain structure dynamics are triggered by long-term disruptions, which may be positive and negative, redesign and implement supply networks in the organization structure, and reconfigure functions in the intelligent structure, redesign of information infrastructure, redesign of supply chain capital flows, product redesign and technical equipment redesign; Supply chain structure dynamics are triggered by interruptions, For example, the elasticity and knock-on effects in the organizational structure are used as a response to supply chain disruptions [81,82]; In the functional structure, the supply chain plan is reconfigured and the information system is reconfigured, unlike the long-term resulting from the strategic level, the tactical level of the structural dynamics emphasize that over a period of time by changing the supply chain to overcome the interruption, and after the interruption, the supply chain structure can return to normal [83,84].

In many cases, supply chain reconfiguration involves the interconnection of decisions at different levels, such as supply chain redesign. Tactical and operational decisions may drive strategic decisions throughout the supply chain structure [85,86]. Supply chain reconfiguration often involves the reconfiguration of supply chain processes and supply chain structures.

Patrucco et al. [87] studied how industry 4.0 technology can support process reengineering in the construction industry context. Tripathi and Gupta [88] proposed a redesigned procurement process combined with various technologies and recommend a redesign of the procurement framework. Dutta et al. [20] examined the research on the use of blockchain technology in supply chains. It is shown that different industries can use blockchain technology to enhance visibility and business process management and to transform with blockchain-based technologies successfully. Grover et al. [89] suggested that the reengineering of processes is complex and involves many factors and that in order to be successful, change must be managed with a balance of attention to all identified factors. The new business process reengineering model is designed to achieve significant performance improvements through a complete redesign of the organization: it replaces

the information system development focused primarily on automating and supporting existing organizational procedures and the process reengineering lifecycle [89].

SC moves from a fixed and statically allocated physical system of processes/products/data to dynamic refactorable services for some enterprises, consisting of different, different, and different elements. SC's ability to reassign and reorganize resources for damage to the organization's external environment can be defined [90]. Sirmon et al. [91] reported that it is also necessary to adjust its resource-based view when an organization is faced with disruption. This points the way for future SC reconfiguration and its precursor research. It should be noted that enterprises with experience of disruption in the past have proactively allocated and adjusted resources [92] and give sufficient time to scan the environment for better response to interrupts [93].

Supply chain reconfiguration includes: supply chain network structure [24,94], contingency plan [12,24,95] and redundancy creation [95,96].

According to Dolgui et al.'s [97] research, reconfigurable SC is a low-cost, responsive, sustainable, and resilient network that is increasingly data-driven, dynamically adaptable, and able to respond to rapid structural changes in physical and cyberspace. Rapidly adjust supply and production capacity and functions to respond to sudden changes by rearranging and reallocating or changing their components.

In order to adapt to changing customer needs and meet the needs of more product customization, many organizations need to reconfigure their manufacturing systems and supply chains. Sabzevari et al. [98] introduced a new, multi-objective, mathematical model for supply chain construction in new products by considering risk management. Al Naimi et al. [99] have studied the causes of resilience and emphasizes the importance of elasticity in reconfiguring the supply chain. The results show that risk management culture, agility, and collaboration positively impact supply chain resilience. New technologies provide solutions that can be more connected, intelligent, autonomous and reconfigurable, and even real-time action [100]. Lou et al. [101] proposed a supply chain reconfiguration method based on adaptive variable neighborhood search (AVNS) to improve the structural robustness of the supply network in the face of random and targeted interruptions. As a disruptive innovation technology, enterprises can use blockchain technology to restructure internal and external resources. When the supply chain is interrupted, there are some new demands. Enterprises can use blockchain technology to obtain the required information and reconfigure or even develop some new supply chains to meet the new demands according to the actual situation of the supply chain. The use of advanced technology to make enterprises more rapid response to changes in the market, and at any time to adjust the enterprise's supply chain. Especially in crisis, enterprises can use blockchain technology to restructure the supply chain, real-time capture changes in market demand according to new decision-making conditions and environment to adjust the supply chain, thereby improving the viability of the supply chain.

### 4.2. External Resource Capacity

The resource dependence theory (RDT) holds that organizational performance depends on its environment and emphasizes inter-organizational efforts to ensure resources and reduce environmental uncertainty [102]. RDT identifies two main organizational goals: minimizing dependence on other organizations and maximizing their dependence on themselves [103]. Unlike the resource-based view, which emphasizes dependence on internal resources, the resource-based view focuses on the resources and capabilities that an organization already owns or may have to build a competitive advantage. RBV provides an examination of the organization's internal resource relationships and capabilities to explain why and how some organizations have outpaced others to gain a competitive advantage [37]. RBV believes that the more added value an organization provides to its customers, the stronger its competitive advantage [104]. The basic principle of RBV is that enterprises have, develop and unique resources and capabilities to create performance advantages for enterprises. If potential capabilities resist imitation through time compression,

uneconomically, historical uniqueness, and causal ambiguity, these advantages are often sustainable [37]. The theory of resource dependence also provides organizations with effective control mechanisms to adjust their structure and behavior to reduce the uncertainty and dependence of blockchain on supply chain time and the external environment. It answers our RQ1. Applying the concepts of organizational effectiveness, interdependence, and external control in resource dependence theory from the perspective of the enterprise value chain will allow enterprises to consider how to obtain resources and capabilities from the external value chain so that enterprises can have and control more power. So from the perspective of resource dependence theory, enterprises must decide how to minimize their dependence on external resources and maximize the dependence of value networks on themselves.

The implementation of blockchain technology in the supply chain can see the improvement of the internal capabilities brought by blockchain implementation to enterprises, and note that the implementation of blockchain technology also needs to rely on external resources and access to external resources. It answers the RQ2 and blockchain technology can help organizations build more robust supply chain capabilities through collaboration, agility, visibility, and more, which are critical to improving supply chain viability in the event of supply chain disruption. For enterprises to implement blockchain technology in the supply chain, it is necessary to provide the internal capacity of the supply chain but also to rely on external resources.

In this study, we primarily use the technology, organizational and environmental (TOE) framework to evaluate the external resources on which the enterprise depends. TOE theory suggests that this progress may be related to factors such as technology, organization, and the environment when firms want to improve their competitive advantage. The TOE framework provides an excellent theoretical perspective for the study of enterprise innovation and technology adoption and is widely used to research of various information systems.

Scholars often use the TOE framework to study the adoption and implementation of specific phenomena. In some resource-based studies, they often use the TOE framework to study the contributing and obstacles to the adoption of specific technologies, such as Fernando et al. [105] used the TOE theory as the basis for the development of the technology adoption framework, examine the drivers of blockchain technology adoption and carbon performance. Caldarelli et al. [106] investigated blockchain technology adoption barriers according to an adjusted TOE framework. The adoption of the TOE framework can be used to assess the drivers and barriers to the motivation for digital transformation [107]. The adoption of blockchain technology depends mainly on senior management support and organizational readiness, and, because of their unique resources and status, large enterprises are more willing to adopt blockchain technology than SMEs.

In the case of supply chain interruption, supply chain viability and become the critical issue of enterprise viability, the traditional research is based on the resource-based view to study the promotion and obstacles of technology adoption in TOE, and few are based on the theory of resource dependence to use the TOE framework to study the enterprise's dependence on external resources. Therefore, from the perspective of resource dependence theory, the framework of TOE needs further development. Therefore, based on resource dependence, this study measures the enterprise's dependence on external resources based on the technical organization and environmental elements in the TOE framework.

## 5. The Relationship between Blockchain Technology and Supply Chain Viability

The world is influenced by demand fluctuations, process uncertainty, supply chain complexity, and information ambiguity. The supply chain is more characterized by VUCA, where, V-volatility, U-uncertainty, C-complexity and A-ambiguity (fuzziness). In particular, as a result of the COVID-19 outbreak, supply chains have become more fragile, uncertain, complex, and ambiguous. Therefore, under the influence of the ongoing COVID-19 Black Swan incident, in such crises, flexibility, agility, and sustainability are no longer the most

critical issues for the supply chain. The essential issue in the supply chain viability, such as for some supply chains, the rapid increase in demand, there will be supply inability to cope, such as masks, hand sanitizer, protective clothing, etc., and for other supply chains, the sharp decline in demand and supply will lead to production shut down or even the risk of bankruptcy, so supply chain viability problems are very significant.

Blockchain increases the availability of resources between virtual partners outside the boundaries of an organization. It conforms to the resource-based view and transaction cost theory [108]. The adoption and diffusion of blockchain technology optimize the internal resources of the enterprise and integrates its business operations and switching partners with greater efficiency and lower transaction costs. Therefore, researchers have the opportunity to extend the focus of these theories from the perspective of organizational resource use to the perspective of inter-organization cooperation and resource sharing. In addition to ensuring data storage and sharing, blockchain can also help synchronize data, coordinate processes, and integrate functions. Many SC use cases have been documented, highlighting the value of BC in driving better operational and competitive performance. This is another important step in developing of digital SC, which can also be seen as an integral part of the "Industry 4.0" phenomenon. The implementation of blockchain technology in the supply chain can significantly impact the viability of the supply chain. The ability of enterprises to rely on external resources can also significantly impact the viability of the supply chain. This answers our RQ3.

*5.1. The Relationship between Blockchain Technical Enabled Supply Chain Capabilities and Supply Chain Viability*

5.1.1. Connectivity and Supply Chain Viability

An essential feature of blockchain-enabled supply chain capability is the connection capability, which refers to the interoperability of technology and the interoperability of organizations by enterprises using blockchain technology. Blockchain in the supply chain area is the most important feature of information sharing, and support information sharing of the underlying technology is the ability to connect. For any organization, the implementation of blockchain technology needs the financial support of enterprises, more importantly, to make different operating software and systems to achieve connectivity. For traditional enterprises, the most basic connectivity technologies such as RFID, enterprise resource planning (ERP) systems, IoT technologies, etc. [109], so that in order to be connected, different applications related to the supply chain need to be connected, and blockchain technology is a tool for storing information, blockchain technology as a guaranteed service to generate reliable information for all supply chain participants. The bottom layer of the technology is the need for blockchain to connect with other devices or applications.

IoT technology and RFID technology can improve internal and external connections between organizations to varying degrees to gain valuable information. When these technologies are integrated with the blockchain platform, the acquisition and dissemination of information become possible [46–48]. At the same time, relying on this technology can increase the visibility of information, so the digitalization of the supply chain can help enterprise decision-making and business integration. Connectivity between upstream and downstream of the supply chain facilitates collaboration between supply chain partners, enhancing supplier upstream and downstream interactions [110,111]. At the same time, the development, implementation, and use of digital technology depend on people to maximize value, regardless of the connectivity required by the firm, in which case the lack of leadership, digital capabilities, support from senior management, and training and support for employees are the main factors affecting the interaction and connectivity between upstream and downstream of the supply chain [111,112]. Especially for blockchain projects in the supply chain, the importance of common collective goals, such as IT investments and risk segments [8]. The tight interconnection between related parties in the supply chain makes the entire system vulnerable to a single outage when the disrupted party cannot handle the crisis [113]. The blockchain-enabled supply chain increased visibility between

supply chain agents to predict possible problems, supply chain disruptions, or possible failures in the enterprise [8].

In the face of severe supply chain disruptions, enterprises must minimize the impact of supply chain disruptions [97]. Information is vital to the viability of the supply chain. If the ability to collect information is not timely, collect the wrong information is related to enterprises' viability. All of this can be achieved through blockchain technology, which increases the visibility of information between supply chain agents [114]. Therefore, the connectivity of the blockchain-enabled supply chain is critical to the supply chain viability of the enterprise and enables the enterprise to adapt and respond quickly and take recovery actions in the event of a severe disruption of the supply chain through interoperability between technologies and interoperability between organizations. The implementation of blockchain technology in the supply chain can minimize the risk, the risk in the supply chain or not lack of characterization, will lead to instability, and through the blockchain technology connectivity, make the information in the supply chain more transparent and promote information sharing, supply chain agents can be based on this real and visible information to better support and manage the risks in the supply chain. The connectivity of the supply chain can be understood on the one hand as a technical resource to facilitate information sharing, but also as organizational capital, a resource derived from information flows, so that by investing in technological resources that can innovate and enhance capacity, the organization can improve its ability to manage risk [115,116].

To sum up, we propose the following:

P1: Blockchain, technology-enabled supply chain connectivity, can improve the supply chain viability.

### 5.1.2. Network Capabilities and Supply Chain Viability

The network capability of the supply chain supported by blockchain refers to the ability of enterprises to establish fully, process, and utilize upstream and downstream enterprise relations by using blockchain technology, and quickly identify market opportunities and respond to market changes by establishing interdependent relationships with upstream and downstream enterprises. The networking ability of an enterprise refers to the ability to establish, process, and utilize relationships, which plays an essential role in building competitive advantage [117], and also the academic defines the network as an individual qualification, i.e., human capital, social qualification, relationship skills or cooperative ability. These terms are associated with communication skills, negotiation skills, the ability to generate trust, conflict management skills, empathy, and a sense of justice. Vesalainen and Hakala [118] identify the roles of network capabilities in the strategic capability architecture of industrial case enterprises. Zacca et al. [119] measured the impact of network capabilities on small business performance through knowledge creation. Network capabilities are dynamic capabilities that generate interdependencies within and outside the organization [67]. Network capabilities enable enterprises to access different resources, identify opportunities, and respond quickly to rapidly changing market demands [76]. Developed network capabilities are the key drivers of the success of central enterprises [120], which is also a critical factor in the performance of large enterprises. Specifically, managing internal and external information flows can improve business performance by facilitating knowledge sharing, downgrading costs, increasing surpluses, and identifying opportunities [69]. Therefore, implementing blockchain technology in the supply chain enhances the network capabilities of enterprises to improve their performance [121].

The network capability of implementing blockchain-enabled supply chain is the ability of the company to share knowledge internally and externally, specifically, employees of enterprises and their upstream and downstream enterprises in the supply chain can use blockchain technology as a hub for information sharing and transmission through the implementation of blockchain technology in the supply chain. For example, store the best organizational practices in blockchain [122]. In this regard, enhanced communication within and outside the enterprise enables the organization to absorb and distribute

knowledge, thereby contributing to the decision-making [123] and enhancing the ability of enterprises to respond to supply chain risks. In addition, network capabilities are developed through blockchain technology to reduce transaction costs [124]. By embedding blockchain technology in the supply chain, making relationships between upstream and downstream businesses more transparent, enterprises reduce opportunism and build trust by accepting mutual monitoring mechanisms [69]. This mechanism is crucial for both small and medium-sized enterprises and large enterprises. Therefore, for the supply chain upstream and downstream enterprises, it also faces misinformation between partners, such as the bullwhip effect and ripple effect. Implementation of blockchain technology in the supply chain can reduce the problem of information asymmetry, and can establish trust between supply chain partners, that is, machine trust, and supply chain enterprises through more fluid, real, equal, and trustworthy interaction, and enhance the enterprise's network capabilities to improve the enterprise's ability to respond to risks.

Effective management of information flows within and outside the supply chain can help organizations identify opportunities and accelerate innovation [70,125]. The ability of enterprises to coordinate internal and external knowledge helps enterprises to identify market trends and quickly adapt to market needs [67]. Therefore, the structured acquisition of heterogeneous knowledge from different sources can enhance the innovation process, thereby ensuring the feasibility of enterprise value proposition and the long-term success of enterprises. At the same time, the network capabilities based on blockchain technology can make supply chain enterprises feel a sense of belonging, which can be active customer identification and gain more status to negotiate with other enterprises. Specifically, embedding blockchain technology in the supply chain can overcome reputational responsibility [69] and enhance the ability of supply chain enterprises to respond to risks while improving supply chain viability. In summary, these arguments demonstrate that the network capability guaranteed by embedded blockchain technology in the supply chain can improve the supply chain's ability to respond to risks and, in turn, enhance the ability to survive.

To sum up, we propose the following:

P2: Blockchain technology enabled the supply chain's network capabilities can improve supply chain viability.

### 5.1.3. Supply Chain Reconfiguration Capacity and Supply Chain Viability

The reorganization capability of the blockchain-enabled supply chain refers to the reorganization of the organizational structure and the reorganization of business processes using blockchain technology in the supply chain. In today's society, many organizations recognize the need to configure their manufacturing systems and supply chains to meet changing customer needs, as well as new product development needs, and making reconfiguration decisions require system-level optimization [126]. In the new decision-making environment, traditional supply chain theories (such as JIT theory, lean production) are no longer adapted to the requirements of the new environment. Therefore, in the increasingly volatile environment, enterprises need to think more about the viability of the supply chain.

In a supply chain disruption event, enterprises can use supply chain resilience to restructure the supply chain, emphasizing the importance of resilience in supply chain reconfiguration [99]. The supply chain is a complex, dynamic network system that changes shape and structure over time.

Blockchain technology-enabled supply chain reconfiguration capabilities, i.e., implementing blockchain technology in the supply chain, can help enterprises implement supply chain reconfiguration by facilitating sharing information between partners and enterprises in the supply chain. Structural changes in the supply chain can produce positive changes based on new disruptive technologies and negative changes due to disruption risks such as natural disasters and ripple effects [127]. Technologies such as big data, cloud computing, and industry 4.0 push for changes in supply chain structural design, i.e., supply chain structure adaptation to new technologies and innovations [128]. Moreover, severe disruption such as a global pandemic (COVID-19) or natural disasters, local wars, etc., may result

in a temporary inability of some suppliers or even large supplier groups to supply rollers forced to change [29]. Dynamic management and optimization of the structure play an important role in determining the competitiveness of enterprises in the market and have an essential impact on the viability of the supply chain. Supply chain reconfiguration tames supply and production capacity and functions by rearranging or reallocating or higher of its components in response to sudden changes. Blockchain-enabled reconfiguration capabilities of the supply chain, including structural diversity, process flexibility, parameter redundancy, and execution visibility. With all four capabilities in place, supply chain resilience can be improved in the event of severe disruption risk.

In summary, we propose the following:

P3: Blockchain technology-enabled supply chain reconfiguration capabilities can improve supply chain viability.

### 5.2. External Resource Dependence and Supply Chain Viability

An enterprise's competitive advantage not only comes from its internal resources and capabilities but also from outside. The resource dependence theory provides us with a new perspective, that is, the organization can establish a competitive advantage from the living environment. The two main organizational goals identified by resource dependency theory are to minimize dependence on other organizations and maximize their dependence on other organizations [103]. Apply the concepts of organizational effectiveness, interdependence, and external control in resource dependence theory to implement blockchain technology in the supply chain.

Resource dependence theory is more important than explaining why more and more organizations are entering into inter-organizational [129,130] and establishing alliances and joint ventures. Enterprises can effectively acquire knowledge and resources between business partners [131], i.e., enterprises that lack specific resources can obtain these resources through the establishment of external relationships, and the theory of resource dependence suggests that enterprises obtain complementary assets by strengthening interdependence [132]. The study found that developing the core competitiveness of an enterprise through knowledge exchange, investment in specific assets and the development of complementary capabilities is a trend [133].

The theory of resource dependence holds that a company's ability to cope with extreme changes in supply and demand will be limited by external entities that control the resources required. [28]. In general, firms try to reduce or improve their forecasting power to reduce their dependence on external resources, but in the case of the COVID-19 pandemic, the stronger the external resource dependence of the enterprise, the more resources it can have, which is more critical to the viability of the supply chain.

In a typical supply chain environment, the structure deployed to manage dependencies may be effective. However, dependencies themselves can evolve. In today's society, organizations rely more or less on external environments [134]. This study relies on the theory of resource dependence to describe the resources and capabilities that organizations may have to rely on their external environment to obtain supply chain viability. Using the TOE theory [135], the resource strength of the enterprise is evaluated. According to the TOE framework, analysis of an enterprise's capabilities shows that when the environmental capability is higher than the technical and organizational capabilities, the enterprise, in general has established a solid external supply chain dynamic relationship. However, in times of severe disruption to the supply chain or when enterprises are facing crisis periods, having greater resource dependence is critical to the viability of the enterprise in times of crisis.

Supply chain viability is determined not only by internal capabilities but more importantly, by internal and external management resources and capacity accumulation [29]. Supply chain viability plays a vital role in supply chain interruption risk. In the supply chain crisis period, consider the complexity of the supply chain and viability ability pay

more attention to the external environment. External resource dependence is an effective way to realize supply chain viability.

For implementing blockchain technology in the supply chain, some enterprises and even large enterprises rely on external partners to develop blockchain applications. Some enterprises are still investing in blockchain technology and building open networks during the crisis, in which case, building their networks gives them more control over their supply chains. When there is much uncertainty, companies want to manage their transaction costs and do not want to invest in certain assets, which could explain why so many companies are searching for outside help to improve their competitive advantage [134]; For the public enterprise, an internal capability is more important than external capability, but in the supply chain scenario, the supply chain itself depends on the connection and interaction of external networks, so it is also imperative to improve the ability of the supply chain to rely on external resources.

In summary, the stronger the dependence on external resources, the more external resources and networks the enterprise has, it is easier for the supply chain to improve the viability of the supply chain. It will correspond to RQ3.

Therefore, we propose the following:

P4: The external resource dependence of supply chain enterprises can improve supply chain viability.

To summarize, the conceptual model proposed is shown in Figure 4:

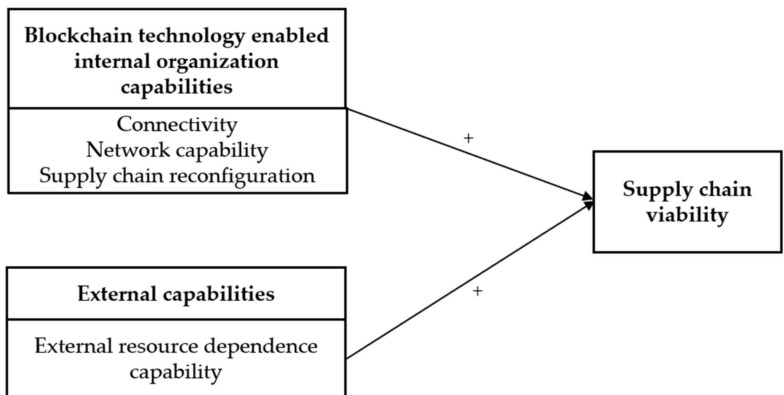

**Figure 4.** Conceptual model of supply chain viability based on blockchain technology.

## 6. Conclusions

The COVID-19 pandemic has provided an opportunity to establish supply chain viability in times of crisis. In severe disruption, whether the implementation of blockchain technology can improve supply chain viability has become a new research hot spot. Based on previous research, this study is based on the concept of the resource-based view and resource dependence theory, the study of blockchain technology-enabled supply chain internal organizational capacity of the specific dimensions and external capabilities of the dimensions and measurement methods, as well as blockchain technical enabled supply chain internal capabilities and external resource dependence how to improve supply chain viability.

This study investigates the blockchain-enabled supply chain's internal capability and external capability based on the resource-based theory and resource dependence theory. All in all, this study points out four propositions, and we posit that the blockchain enables the supply chain capabilities: connectivity, network capabilities, and supply chain reconfiguration capabilities will have a positive impact on the supply chain viability and the external resource dependence of supply chain enterprises can improve supply chain viability. These findings will help the supply chain manager invest in technology to improve the ability to cope with risk. Moreover, these findings will improve the blockchain-enabled supply chain internal and external capability to form the supply chain viability.

### 6.1. Theoretical Contributions

In general, enterprises often face different barriers to adopting new technologies, but during crises such as the COVID-19 pandemic, new opportunities are offered to implement new technologies. In the crisis environment, the decision-making environment and conditions of enterprises have significantly changed. In contrast, in profound uncertainty, the crisis may force enterprises to seek new technologies to improve their ability to deal with the crisis and risk.

First, in the supply chain environment, the perspective provided by resource dependence theory may be better than that provided by the resource-based view. The adoption of the TOE framework is also an effective way to evaluate resource dependence capabilities.

Second, based on exploratory research, we put forward four general research propositions, we believe that the internal organizational capacity of blockchain technical support and external resource dependence ability can improve the supply chain viability of enterprises, the combination of internal and external capabilities, this joint perspective of cumulative capacity also needs to be more in-depth research.

Third, in the face of supply chain risks, organizations may be more willing to invest in new technologies to improve supply chain viability and blockchain as a disruptive innovation of new technologies. Especially in crises, it provides some new opportunities for enterprises to adopt new technologies and requires in-depth research on implementing blockchain technical support in the supply chain.

Fourth, organizations should use the COVID-19 incident as an opportunity to test and rectification internal and external capabilities [136]. In severe supply chain interruption, the new research problem has gone beyond the scope of the traditional theory of agility, resilience, lean production, and so on. In the case of a truly broken supply chain, Traditional supply chains are no longer able to meet the current needs, and new supply chain networks need to be redeveloped to cope with the new decision-making environment. Therefore, how to use blockchain technology and its ability to improve the viability of supply chain is also a hot issue in future research under the current increasingly turbulent environment.

### 6.2. Practical Contributions

The COVID-19 pandemic has brought more complex and realistic problems to supply chain managers, especially in the current supply chain network's more complex and more fragile environment. Supply chain managers need to deal with short-term and long-term crises and take measures from the technical, organizational, and environmental levels to deal with the crisis. This study can provide some valuable references for supply chain managers.

First, this study helps supply chain managers to take technical investment, especially in the COVID-19 crisis. Enterprises that improve the viability of the supply chain need to manage the supply chain effectively, and adopting new technology is a crucial way to manage the supply chain effectively.

Second, in a crisis environment, improving supply chain viability requires investment in resources and capabilities within the organization and investment in external organizational relationships and partnerships, especially for supply chain enterprises. External resource dependence capacity may be more important than the internal capacity of the organization.

Third, according to this study, enterprises can see crisis events as opportunities for innovation and improvement [137]. During a crisis, enterprises may face less pressure to implement new technologies, and adopting new technologies during a crisis may be seen as an opportunity to overtake on a bend or find breakthroughs for businesses.

Finally, enterprises should pay attention to the supply chain capacity endowed by blockchain technology and combine the external resources they need. Therefore, for managers, Managers can use the perspective of combining RBV and RDT theories proposed in this study and the TOE framework to identify the internal organizational capabilities and external resource dependence capabilities required by enterprises.

*6.3. Research Limitations and Future Research Directions*

This study provides a joint perspective based on the resource-based view and resource dependence theory and provides new research insights for managers. However, the limitation of this study lies in that the propositions put forward are not empirically verified, so one of the directions of future research is to use more data to verify the propositions presented by empirical analysis.

This study is also the first to combine blockchain technology in supply chain viability with the earliest research of enterprises. At present, supply chain viability is an emerging research trend. In the future, research on the relationship between blockchain and supply chain viability can be strengthened and how to use new technologies to improve supply chain viability.

All in all, the past decades have seen the emergence of abundant supply chain theories. The crisis brought by the COVID-19 pandemic has also brought a series of new challenges to the supply chain field, thus giving rise to a new supply chain research field: supply chain viability [29]. In times of crisis, enterprises are more inclined to use new technology to solve the problems they may face. Therefore, in a new decision-making environment, this paper studies how to use blockchain technology to enhance the viability of supply chain enterprises has an efficient significance and theoretical significance, but also provides some guidance to improve the supply chain viability for supply chain practitioners.

**Author Contributions:** W.Y. and W.R. contributed equally to this article. All authors have read and agreed to the published version of the manuscript.

**Funding:** This research was supported by the National Natural Science Foundation of China of Projects No. 71661029.

**Institutional Review Board Statement:** Not applicable.

**Informed Consent Statement:** Not applicable.

**Data Availability Statement:** The data presented in this study are available on request from the corresponding author. The data are not publicly available due to [the enterprise which provides the price data are reluctant to disclose their identities].

**Acknowledgments:** This work was supported by the National Natural Science Foundation of China of Projects No. 71661029.

**Conflicts of Interest:** The authors declare no conflict of interest.

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
