# Peer review of "Theoretical Exploration of Supply Chain Viability Utilizing Blockchain Technology"

_sustainability, doi:10.3390/su13158231_

Round 1

Reviewer 1 Report

The text is very interesting from the point of view of the current market direction, which, however, could be shortened and the main ideas of the authors could be selected.

First of all, it is necessary to discuss what kind of text it is - if it is a theoretical search, it is also necessary to state it within the type of contribution.

At the beginning of the articles, it would be necessary to identify the methodology - how the following data (theoretical research) were processed, evaluated and synthesized into the final form presented.

At the beginning of the article mentioned problem areas, but what questions, hypotheses of the article to answer - highlight in the introduction of the article.

The text of the article is quite dense in the final article, it would be appropriate to highlight the main research questions and answer them briefly - to create the main results in a simplified form.

Author Response

Thank you so much for the helpful feedback. We've made changes in accordance with your specific requests.

Point 1: The text is very interesting from the point of view of the current market direction, which, however, could be shortened and the main ideas of the authors could be selected.

Response 1: The article has been significantly condensed.

Point 2: First of all, it is necessary to discuss what kind of text it is - if it is a theoretical search, it is also necessary to state it within the type of contribution.

Response 2. We explicitly emphasized in the amended version's abstract and body that this is a conceptual theoretical research paper and the contribution of this study.

Point 3: At the beginning of the articles, it would be necessary to identify the methodology - how the following data (theoretical research) were processed, evaluated and synthesized into the final form presented.

Response 3. A method chapter is included to the main body of the article to explain how the literature is chosen.

Point 4: At the beginning of the article mentioned problem areas, but what questions, hypotheses of the article to answer - highlight in the introduction of the article.

Response 4. We match the three questions in the introduction to the propositions stated in the revised version.

Point 5: The text of the article is quite dense in the final article, it would be appropriate to highlight the main research questions and answer them briefly - to create the main results in a simplified form.

Response 5: We have trimmed and polished the last chapter in the main body to highlight the important research topics and research contributions.

Author Response

Thank you so much for the helpful feedback. We've made changes in accordance with your specific requests.

Point 1: Overall, though the literature collection of this study is complete, but there is no description of the methodology. This will challenge and question the Propositions proposed in the paper.

Response 1: To address this question, we introduce a brief description of the methodology in the chapter 3.

Point 2: In my opinion, I think the English writing of this article needs a lot of improvement. There are many sentences in the text that are too long to be readable. Further, some sentence descriptions are incomplete and some sentences use improper punctuation.

Response 2. I've invited an international student to assist me in revising the paper's English writing level. If I need to rewrite it further, I'll do so with the help of a professional polishing service.

Point 3:  I think that not only Section 2, but also Section 3 is like a literature review.

Response 3. The second chapter is a literature review, and the third chapter is a discussion of the important supply chain capabilities enabled by blockchain, which we refined based on the review. Because we believe this topic is conceptual, a detailed exposition in the theoretical section is required, and in response to your suggestion, I have simplified the third section.

Point 4: Some of the cited references of the literature such as Croom et al. (2007), Wang and Lo (2016), Tyler (2001) and Zhu et al. (2018) are missing. Please check.

Response 4. The paper's missing documents have been added where needed.

Point 5: The concepts to be stated in Figure 1 are not specific enough enough to correspond to the proposed Propositions.

Response 5. The contents of Figure 1 have been adjusted, and the above-mentioned assumptions can be modified as well.

Point 6: When some abbreviations such as UTAUT, ICT, BC in the text appear for the first time, please display their full names.

Response 6: We have added the complete names of any abbreviations that appear in the article for the first time.

Reviewer 3 Report

Authors provide an interesting survey on the adoption of blockchain in supply chain-oriented scenarios, thus providing a huge amount of concepts and details on how this paradigm can be applied. Nevertheless, even if the proposed paper is a concept paper, readers may be bored in reading such a paper without any figure before the concluding section. Authors are required to better contextualise their concepts with the aid of figures and illustrations, otherwise such a theoretical contribution may be misunderstood.

Finally, authors should carefully verify and check their paper, since it contains a lot of typos and errors that should be sanitised before the final publication.

Author Response

Thank you so much for the helpful feedback. We've made changes in accordance with your specific requests.

Point 1: Authors provide an interesting survey on the adoption of blockchain in supply chain-oriented scenarios, thus providing a huge amount of concepts and details on how this paradigm can be applied. Nevertheless, even if the proposed paper is a concept paper, readers may be bored in reading such a paper without any figure before the concluding section. Authors are required to better contextualise their concepts with the aid of figures and illustrations, otherwise such a theoretical contribution may be misunderstood.

Response 1:. To make the article less boring, we incorporated some relevant graphics and tables.

Point 2: Finally, authors should carefully verify and check their paper, since it contains a lot of typos and errors that should be sanitised before the final publication.

Response :2:. I sought the assistance of an international student to help me improve the grammatical section of the article. If changes are still required, we shall seek the services of a specialized polishing company.

Reviewer 4 Report

This study presents the theory exploration on research on supply chain viability based on blockchain technology. I have the following major concerns regarding this study.

-  The abstract is very long and not properly organized, and I hardly found meaningful information that shows the study objective and key findings.

-   Though this study is a kind of review paper authors did explicitly mention the goals/objectives/aims of this research anywhere in the paper

  •   My major concern is regarding the research methodology of this study. Readers cannot learn about
  • the need for this study,
  • what are the research questions answered through several papers
  • How were selected papers identified for this study?
  • What is the quality of those papers, and how they assess the quality of the selected paper?
  • Another major concern is regarding the presentation of the paper. Currently, this paper has VERY LONG SECTIONS that do not deliver meaningful messages to the reader. For example, Section 3.1.1 and many other

Significant writing improvement is required. In many places’ sentences are very long. For example, (“From the perspective of the resource-based view and resource-de-pendent theory, this study studies the specific dimensions of the supply chain capability enabled by blockchain technology and the impact of external resource-dependent capability on the viability of the supply chain, and theoretically, studies how internal organizational capability and external re-source-dependent capability affect the supply chain viability”) in the Abstract.

Author Response

Thank you so much for the helpful feedback. We've made changes in accordance with your specific requests.

Point 1: The abstract is very long and not properly organized, and I hardly found meaningful information that shows the study objective and key findings.

Response 1. The abstract section has been condensed and revised based on your suggestions.

Point 2: Though this study is a kind of review paper authors did explicitly mention the goals/objectives/aims of this research anywhere in the paper

Response 2. Because this is a conceptual study, we also state our research goals and objectives, which can help to clarify our theoretical contribution.

Point 3: My major concern is regarding the research methodology of this study. Readers cannot learn about the need for this study, what are the research questions answered through several papers; How were selected papers identified for this study? What is the quality of those papers, and how they assess the quality of the selected paper?

Response 3. To compensate for the lack of method in this work, we added a brief chapter on methodology which shows details in chapter 3.

Point 4: Another major concern is regarding the presentation of the paper. Currently, this paper has VERY LONG SECTIONS that do not deliver meaningful messages to the reader. For example, Section 3.1.1 and many other

Significant writing improvement is required. In many places’ sentences are very long. For example, (“From the perspective of the resource-based view and resource-de-pendent theory, this study studies the specific dimensions of the supply chain capability enabled by blockchain technology and the impact of external resource-dependent capability on the viability of the supply chain, and theoretically, studies how internal organizational capability and external re-source-dependent capability affect the supply chain viability”) in the Abstract.

Response 4. In terms of the paper's statement section, we've cut a lot of space and sought the support of an international student to help us correct the grammar. If we need more editing, we'll hire a professional polishing service to help us enhance our English.

Round 2

Reviewer 2 Report

The authors seem to have responded to most of my earlier comments. I only have several questions and suggestions as follows.

  1. In my opinion, the English writing of this article still need to be improved and there are still many typos and confusing sentences that need to be rephrased as follows:
  • Abstract, line 8, “investigate should be “investigates”.
  • Page 2, Page 27, “which ” should be “Which”.
  • Page 3, line 20, please delete the comma after “[19] ”.
  • Page 4, lines 11 and 17, “examines ” should be “examine”.
  • Page 4, line 14, “proposed ” should be “propose”.
  • Page 4, line 23, “studies ” should be “study”.
  • Page 4, line↑14, “studied ” should be “study”.
  • Page 5, line↑16, “In times ” should be “in times”.
  • Page 6, line↑, what does “between the years of t and 2021” mean?
  • Page 7, line 3, “table 1” should be “Table 1”.
  • Page 7, line↑10, “figure 2” should be “Figure 2”.
  • Page 11, line↑12, “studies” should be “study”.
  • Page 11, line↑9, “examines” should be “examine”.
  • Page 11, line↑5, “suggests” should be “suggest”.
  • Page 12, line7, “reports” should be “report”.
  • Page 18, line↑1, should “figure 3” be “Figure 4”?
  1. Are the researches in Table 1 listed in the References section?
  2. I still think that the concepts to be stated in Figure 4 are not specific enough to correspond to the proposed Propositions. It cannot be seen whether they are positive or negative impacts on supply chain viability. In addition, the figure seems to be repeated. Please check.

Author Response

Thank you so much for the helpful feedback from the reviewers. I've made the following changes based on your suggestions: 1. I've resolved the grammar issues in the paper based on your comments, and the grammar of the article has been deeply modified. As for the problems in Page 6, what I express is to search the literature so far, I have revised the unclear sentences in the article. 2. The references listed in Table 1 have been included. 3. I added a plus sign to the corresponding path for the questions relating to the hypothesis in Figure 4. The plus symbol denotes that the antecedent has a positive impact on the outcome. Thank you very much!

Reviewer 4 Report

Authors have improved the papers according to my pervious comments. Paper may be accepted. One suggestion: When responding to reviewers’ comments please provide the detail response for example, how and where you address comments (e.g., page number, paragraph number, subsection).

Author Response

Thank you for the reviewer's comments, which will be extremely useful for my future submissions and paper publishing. I will pay close attention to your comments in the future submission process, respond to reviewer comments carefully and clearly, and clearly match reviewer comments with my own responses. Thank you very much!